# Characterisation of the *ARF* Gene Family in Salicaceae and Functional Analysis of *PeARF18* in Heteromorphic Leaf Development of *Populus euphratica*

**DOI:** 10.3390/ijms27010335

**Published:** 2025-12-28

**Authors:** Tongrui Song, Hongyan Jin, Jing Li, Qi Ning, Donghui Miao, Yidan Yang, Zhibin Cui, Zhijun Li, Zhihua Wu, Peipei Jiao

**Affiliations:** 1Xinjiang Production and Construction Corps Key Laboratory of Protection and Utilization of Biological Resources in Tarim Basin, College of Life Science & Technology, Tarim University, Alar 843300, China; songtr20001209@163.com (T.S.); j2601803054@163.com (H.J.); jing926819729@163.com (J.L.); ningqi20000423@163.com (Q.N.); mdh1216g@163.com (D.M.); yyd202510@163.com (Y.Y.); 17399975214@163.com (Z.C.); lizhijun0202@126.com (Z.L.); 2College of Life Sciences, Zhejiang Normal University, Jinhua 321004, China

**Keywords:** Salicaceae, *Populus euphratica*, auxin response factors, leaf shape, regulation

## Abstract

Auxin plays a crucial role in plant growth and development via concentration gradient regulation, with auxin response factors (ARFs) as key transcription factors in its signalling pathway. However, comprehensive identification and characterisation of *ARF* genes in Salicaceae remain limited. This study performed a genome-wide analysis of *ARF* genes in three Salicaceae species (*Populus euphratica* Oliv., *Populus pruinosa*, and *Salix sinopurpurea*), aiming to clarify their physicochemical properties, evolutionary relationships, and functional relevance. A total of 34 *ARF* genes were identified in each species, all being nucleus-localised hydrophilic unstable proteins clustered into six phylogenetic subgroups. Their promoters contain numerous *cis*-acting elements responsive to light, phytohormones, and stresses. Transcriptome and qRT-PCR data showed significant up-regulation of *PeARF18* in ovate/broad-ovate leaves of *P. euphratica* compared to linear/lanceolate leaves. This study provides preliminary insights into the characterisation and potential role of the Salicaceae *ARF* gene family, laying a foundation for further functional exploration of *PeARF18* in *P. euphratica* leaf shape development.

## 1. Introduction

Indole-3-acetic acid (IAA), as the main type of auxin, plays a crucial role in plant growth and development in response to cellular and environmental signals [1,2]. Most of these auxin functions are achieved through the regulation of gene expression by auxin response factor (ARF) proteins, which translate IAA signals into gene expression with the help of AUX/IAA proteins [3]. Most ARF proteins usually consist of three conserved structural domains [4]: (1) the N-terminal B3-type DNA-binding domain (DBD), which is responsible for binding to auxin response elements (AREs); (2) a variable middle region, which serves as an activating structural domain or repressor structural domain and determines whether the ARF protein is an activator or repressor [5,6]; and (3) the C terminal dimerisation domain (CTD), which is responsible for protein–protein interactions such as homodimerisation of ARF proteins or heterodimerisation of ARF with AUX/IAA proteins [7]. Notably, some ARF proteins contain only some of these three conserved structural domains. For example, *ARF3*, *ARF13*, and *ARF17* in *Arabidopsis* [8] lack a CTD, and *ARF23* only consists of a truncated DBD.

With the increasing number of plant genomes being sequenced, the *ARF* gene family has been studied at the genome-wide level in various plant species, including *Oryza sativa* L. [9], *Citrus sinensis* [10], and *Populus trichocarpa* [11]. These previous studies have shown that the structure of the proteins encoded by ARF varies considerably within or between families in different species. For example, only one ARF with a truncated DBD was found in *citrus* compared with *Arabidopsis thaliana* [12], whereas a large number of ARFs lacking CTDs were found in rice, maize [13], banana [14], and alfalfa [15]. Thus, the *ARF* gene families in different plant species show great diversity and abundance, most of which need to be further studied and characterised for enhanced understanding.

Substantial evidence indicates that the *ARF* gene participates in multiple auxin-dependent developmental processes in plants [14,16]. For example, *AtARF2, AtARF3, AtARF4*, and *AtARF5* are essential for the development of female and male gametophytes, respectively [17]. *AtARF8* regulates stamen elongation and lignification of the inner wall of the anther chamber [18], whereas *AtARF3* plays a unique role in early flower development [19]. *AtARF7* and *AtARF19* are essential for auxin-mediated plant development through the regulation of a set of unique and partially overlapping target genes [12]. Notably, *ARF* genes also play a crucial role in auxin-mediated leaf development, which primarily encompasses the growth of leaf blades and the formation and differentiation of leaf primordia within the stem apical meristem (SAM). *AtARF3* and *AtARF4* affect leaf polarity development in *A*. *thaliana* [20,21], and *AtARF10* and *AtARF16* are involved in the regulation of *Arabidopsis* leaf growth and development [22]. *OsARF19* is strongly expressed in leaf nodes and involved in BR regulation of rice leaf angle [23]. The expression levels of many *ARF* genes are altered under various abiotic stresses such as drought [24,25] and salt stress [26,27,28]. Additionally, various transcription factors, miRNAs, and growth factors are also essential for this process of auxin-mediated leaf development [29,30]. These studies provide fundamental information for the subsequent characterisation of the function of *ARF* genes in response to cellular signalling and environmental stresses during plant growth and development.

*P.euphratica* is the only naturally occurring tree species in the desert regions of northwest China [31,32]. *ARFs* have been identified in *P. euphratica*, and 34 members of *PeARFs* have been identified [33]. *Populus pruinose* and *Salix sinopurpurea* are both drought-tolerant species within the Salicaceae family [34,35]. These studies indicate that numerous auxin response factors (ARFs) influence leaf traits in plants, whilst in *P. euphratica*, the auxin signalling pathway involving *ARFs* has been found to be associated with heterophyllous leaves [36]. Previous research indicates that, although *ARF* genes have been identified across multiple species and functional research has progressively deepened, the size of the *ARF* family significantly varies between species, and its spatiotemporal expression patterns are complex. Belonging to the drought-tolerant Salicaceae family, compared to the lanceolate leaves of *Salix sinopurpurea* and the hairy broad leaves of *Populus pruinose*, *P. euphratica* possesses four distinct leaf shapes. Why *P. euphratica* has heteromorphic leaves compared with other Salicaceae species, how *ARFs* evolved, and how *ARFs* affect the leaf shape of *P. euphratica* have not been studied.

In this study, we employed bioinformatics approaches to identify and characterise the *ARF* gene family comprising *Populus euphratica*, *Populus pruinose*, and *Salix sinopurpurea* at the genome-wide level. Comparisons among the three species reveal differences of this family between *P. euphratica* and those of the other two Salicaceae species, including motif presence, conserved domains, the number and distribution of *cis*-acting elements, and the proportion of various types of protein secondary structures. Further analysis of subcellular localisation and expression profile of *PeARF18* was performed in four kinds of heteromorphic leaves in *P. euphratica*. This study not only contributes to a comprehensive understanding of the biological properties of the *PeARF* family and the potential role of *PeARF18* in the development of *P. euphratica* heteromorphic leaves for the first time but also provides new gene resources for further conservation and utilisation of the desert tree.

## 2. Results

### 2.1. Identification of ARF Family Members and Prediction of Physicochemical Properties in Three Salicaceae Species

A total of 102 ARFs were identified in the genomes of three Salicaceae species: 34 genes in *Populus euphratica* Oliv., 34 genes in *Populus pruinose*, and 34 genes in *Salix sinopurpurea*, respectively. Predictive analysis of the physicochemical properties of ARF family members from the three species revealed (Appendix A) that the length of PeARF proteins ranged from 395 amino acids (PeuTF06G00760.1) to 1149 amino acids (PeuTF06G01378.1). The molecular weights ranged from 43.85 kDa (PeuTF106G00760.1) to 128.29 kDa (PeuTF06G01378.1), and the pI values of the PeARF proteins ranged from 5.35 (PeuTF05G02165.1) to 9.07 (PeuTF106G00760.1). The lengths of the PpARF proteins ranged from 481 amino acids (PprTF03G1385.1) to 1149 amino acids (PeuTF03G1199.1). The molecular weights ranged from 54.58 kDa (PprTF03G1385.1) to 125.92 kDa (PeuTF03G1199.1), and the pI values of the PpARF proteins ranged from 5.36 (PprTF05G1918.1) to 8.85 (PprTF04G1680.1). The length of the SpuARF proteins ranged from 592 amino acids (SpuTF05G133500.1) to 1136 amino acids (SpuTF06G114200.1). The molecular weight ranged from 65.48 kDa (SpuTF05G133500.1) to 127.19 kDa (SpuTF06G114200.1), and the pI values of the SpuARF proteins ranged from 5.2 (SpuTF05G189800.1) to 8.41 (SpuTF16G085100.1). The PeARFs had the greatest differences in the number of amino acids, molecular weight, and isoelectric point amongst the studied Salicaceae species. A total of 102 members of the ARF protein members had negative mean coefficients of hydrophobicity, indicating that the majority of the *ARF* family are hydrophilic proteins. In addition, subcellular localisation prediction showed that all predicted locations of ARFs were within the nucleus (Appendix A), indicating that all members of the *ARF* family functioned in the nucleus.

### 2.2. Systematic Evolutionary Analysis of the Multi-Species ARF Gene Family

With *Arabidopsis thaliana* as the outgroup, the phylogenetic tree of *ARF* genes in the three Salicaceae plants was inferred (Figure 1). The analysis confirmed the classification of ARF proteins into six subgroups, denoted as groups 1–6. Specifically, group 1 contained 26 ARFs (three AtARFs, seven PeARFs, eight PpARFs, and eight SpuARFs), group 2 contained 20 ARFs (three AtARFs, six PeARFs, six PpARFs, and five SpuARFs), group 3 contained 20 ARFs (two AtARFs, six PeARFs, six PpARFs, and six SpuARFs), group 4 contained 12 ARFs (two AtARFs, four PeARFs, three PpARFs, and three SpuARFs), group 5 contained eight ARFs (eight AtARFs), and group 6 contained 39 ARFs (five AtARFs, 11 PeARFs, 11 PpARFs, and 12 SpuARFs). The number of AtARFs in groups 1–6 significantly differed from that of the Salicaceae family ARFs, and the *ARF* gene family of Salicaceae plants had significant expansion in comparison with that of *Arabidopsis thaliana*. However, no significant expansion was found in comparison amongst Salicaceae plants. Meanwhile, AtARFs underwent aggregation in group 5.

### 2.3. Analysis of Conserved Motifs and Conserved Structural Domains of ARF Family Members in Three Salicaceae Species

In the identification of motifs by MEME analysis, 10 motifs (motif 1–motif 10) were identified for the *ARF* gene family, and the motif module analysis showed that not all *ARF* members contain these conserved motifs, suggesting that different *ARFs* may have different regulatory functions (Figure 2). Motifs 1, 2, 3, and 5 constitute the B3 structural domain, and motifs 8, 9, and 10 constitute the auxin-responsive structural domain. Conserved domain analysis showed that all *ARF* gene family proteins, except *PeuTF03G01613.1* without motif 1, have B3 and auxin-responsive domains, and some *ARF*s have AUX-IAA superfamily domains, which mediate the normal degradation of AUX-IAA and maintain the stability of auxin signalling; multiple sequence alignments also corroborate this finding (Appendix A). By contrast, the extra cond-enzymes-superfamily structural domains in *PprTF01G0727.1* and *PprTF03G1199.1* and the extra PRK12757 superfamily kinase-associated structural domains in *SpuTF18G113500.1* may be the result of adaptation to the environment. Moreover, *PeuTF04G02163.1* and *PeuTF03G00009.1* have a small-fragment B3 structural domain, and both of them were missing motif 7, which may have been fragment-deficient in evolution and affect the function of *PeuTF04G02163.1* and *PeuTF03G00009.1*.

### 2.4. Prediction of Cis-Acting Elements of ARF Family Members in Three Salicaceae Species

The promoter sequence of the 2000 bp region upstream of the initial codon in *ARF* was explored to analyse the biological processes in which *ARF* family members are involved (Figure 3). The results revealed different distributions of *cis*-acting elements amongst family members, mainly divided into four major categories: plant growth and development-related elements, hormone response elements, light response elements, and stress response elements. Amongst them, stress-responsive *cis*-acting elements were mainly involved in defence stress (such as the wound element), drought stress response (such as the AREB element), and hypoxia and low-temperature responses. Phytohormone response elements were mainly the salicylic acid response element (such as the TCA element), the gibberellin response element (such as P-box and TATC-box), the methyl jasmonate response element (such as CGTCA-motif and TGACG-motif), the abscisic acid response element, and the AREs (such as the TGA-element and auxin element). Light-responsive elements were the most abundant, followed by hormone-responsive elements. These *cis*-acting elements were involved in developmental growth, cellular processes, hormone-mediated regulation, and various biotic and abiotic-mediated stresses in plants.

### 2.5. Chromosomal Localisation of the ARF Gene Family in Three Salicaceae Species

The location of the 102 *ARF* genes in the genome was investigated to study the chromosomal localisation of *ARFs* in three Salicaceae species. Analysis of chromosomal location showed that *PeARFs* were distributed on 15 of the 19 chromosomes of *P. euphratica*, with the highest number of *PeARFs* on Chr04 with five *PeARF* members, followed by Chr02 with four *PeARF* members. No *PeARFs* were found on Chr07, Chr08, Chr13, and Chr19. *PpARFs* were distributed on 16 of the 19 chromosomes of *Populus pruinosa*, with Chr02 having the highest number of *PpARFs* with four *PpARF* members, followed by Chr01, Chr03, Chr04, Chr05, and Chr06 with three *PpARF* members. No distribution of *PpARFs* was found in Chr19. *SpuARFs* were distributed on 16 of the 19 chromosomes of *Salix sinopurpurea*, amongst which Chr02 had the highest number of *SpuARFs*, with four *SpuARF* members, followed by Chr01, Chr03, Chr04, Chr05, and Chr06 with three *SpuARF* members. No distribution of *SpuARFs* was noted on Chr19. The *ARF* family members were unevenly distributed on all chromosomes. This uneven distribution on chromosomes suggests that each chromosome contributed differently to the evolution of the *ARF* family. The chromosomal distributions of *SpuARFs* and *PpARFs* were very similar (Figure 4 and Appendix A).

### 2.6. Analysis of Covariance in the ARF Family in Three Salicaceae Species

The covariance of *ARF* between *Populus euphratica* and two other Salicaceae species (*P. pruinosa* and *Salix sinopurpurea*) and *A. thaliana* was compared to further explore the potential evolutionary process of *PeARF* in *P. euphratica* (Figure 5). The results showed 82, 90, and 20 *ARF* covariance pairs between *P. euphratica* and *P. pruinosa*, *S. sinopurpurea*, and *A. thaliana*, respectively, suggesting that the covariance of *ARF* within the Salicaceae species is more conserved than that amongst *P. euphratica*, *P. pruinosa*, and *A. thaliana* (Appendix A). By contrast, *A. thaliana* and *P. euphratica* had only 20 colinear pairs, suggesting that the *ARFs* in Salicaceae species have undergone gene family expansion during evolution.

### 2.7. Predicted Protein Secondary Structures of ARFs from Three Salicaceae Species

The secondary structures of ARF proteins were predicted (Figure 6 and Appendix A); results show that ARF proteins contained four secondary structures: α-helix, β-turn, extended strand, and random curl. The proportion of the secondary structures of ARF proteins is basically as follows: random curl > α-helix > extended strand > β-turn. α-Helix and random curl are the major conformations of ARF proteins. The average percentages of the protein secondary structures of PeARF family members were 14.48% α-helix, 2.31% β-turn, 72.07% random curl, and 11.51% extended strand. The average percentages of the protein secondary structures of PpARF family members were 14.93% α-helix, 2.44% β-turn, 71.42% random curl, and 11.21% extended strand. The average percentages of the protein secondary structure of SpuARF family members were 15.18% α-helix, 2.36% β-turn, 71.46% random curl, and 11.01% extended strand (Appendix A). The PeARF family members had the largest percentage of extended strand and random curl and the smallest percentage of α-helix and β-turn. The SpuARF family members had the largest percentage of α-helix and β-turn and the smallest percentage of extended strand. Such differences may influence ARFs to perform functions differently within different species.

### 2.8. Expression Patterns of PeARFs in Populus euphratica Heterophyllous Leaves

The transcript levels of *PeARFs* across four leaf morphologies in *P. euphratica* were analysed to investigate the potential role of *PeARFs* in the development of heteromorphic leaves in *P. euphratica.* The expression data for *PeARF* gene family members were extracted across these four leaf morphologies and visualised using TBtools. The expression patterns were primarily categorised into two types: up-regulation and down-regulation in ovate (Ov) and broadly ovate leaves (Bo) (Figure 7A). Multiple *PeARFs* were previously identified through multi-omics integration analysis and molecular regulatory network analysis. qRT-PCR further validated the *PeARF* expression results, which were consistent with the results of transcriptomic analysis. The correlation coefficients between transcriptomic data and qRT-PCR data are as follows: (PeuTF14G00975.1: r = 0.999, PeuTF15G01074.1: r = 0.524, PeuTF01G00853.1: r = 0.941, PeuTF02G00687.1: r = 0.986, PeuTF04G00423.1: r = 0.777, PeuTF18G00580.1: 0.607). qRT-PCR results indicate that the selected genes align with the transcriptome expression pattern, with *PeuTF01G00853.1*, *PeuTF04G00423.1*, and *PeuTF14G00975.1* exhibiting significantly elevated expression levels in broad leaves (Ov and Bo) and *PeuTF02G00687.1* exhibiting significantly reduced expression in broad leaves. Although *PeuTF18G00580.1* and *PeuTF15G01074.1* showed no significant differences between narrow leaves (Li and La) and broad leaves, their expression levels still demonstrated an increasing and decreasing trend, respectively (Figure 7B). Notably, among them, *PeARF18* (*PeuTF14G00975.1*) exhibited the highest expression in Ov and Bo leaves, compared with the expression level observed in Li and La leaves. Integrating with previous reports on *ARFs*‘ involvement in leaf morphogenesis, *PeARF18* may be involved in regulating the development of *P. euphratica* heterophyllous leaves.

### 2.9. Subcellular Localisation Analysis of PeARF18

The function of a protein is generally determined by its localisation within the cell. Once mature proteins are localised to specific organelles, they can perform their biological functions. By using transient transformation technology in tobacco, subcellular localisation of PeARF18 was observed under a laser confocal microscope with excitation at 514 nm and detection at 524 nm. The results revealed distinct fluorescence in the cell nuclei, indicating that PeARF18 functions as a transcription factor within the nucleus (Figure 8).

## 3. Discussion

### 3.1. Identification of Three Salicaceae ARF Family Members

*P. euphratica* has evolved a unique heteromorphic leaf system to effectively cope with the arid desert environment. The structural features of the heteromorphic leaves enable *P. euphratica* to effectively retain water, increase photosynthesis, reduce transpiration, and enhance tolerance to drought stress [37]. Research indicates that *ARFs* play a crucial role in plant growth and development. The *AtARF* family was firstly identified with 23 members. The *ARF* family of common botanical model plants have been identified one after another, and their functions and mechanisms of action have been studied in depth.

Comparative analysis of the *ARF* family across numerous species revealed homologous relationships amongst *ARF* members within the same species, and *ARF* members from different species exhibited a degree of homology. The phylogenetic analysis of apple *ARF* genes revealed their division into three groups [38]. Subsequent phylogenetic tree analysis and subgroup classification of apple *ARF* genes, alongside those from *Arabidopsis*, rice, tomato, maize, and grape, further categorised 148 *ARF* members into five distinct groups. Fifty *ARF* genes identified in tobacco formed 22 paralogous gene pairs, with 20 pairs exhibiting highly similar phylogenetic relationships and gene sequences. Based on the phylogenetic tree of barley, maize, rice, and *AtARF* genes, 96 *ARF* genes from the four plants were grouped into six clusters (I–VI). The findings in *Arabidopsis* were similar [39]. The phylogenetic analysis of *ARF* families from *Arabidopsis*, *P. euphratica*, *P. pruinosa*, and *Salix sinopurpurea* showed that group 5 was the *Arabidopsis* genes that underwent clustering, suggesting that they have a specific role in *Arabidopsis*. By contrast, all other groups underwent gene expansion, which may allow them to more finely differentiate their functions and regulate plant growth and development; this may play a crucial role in shaping the structure and evolution of species genomes.

### 3.2. The Role of ARFs in Plant Growth and Development and Responses to Abiotic Stress in Three Salicaceae Species

Variations in gene expression levels are largely determined by *cis*-acting elements within the promoter region of the genes, which provide the essential molecular basis for responding to environmental stimuli. Analysis of *cis*-acting elements in *ARF* gene promoters elucidates their potential regulatory responses to abiotic stress and plant growth and development. In the absence of auxin, ARF interacts with AUX/IAA proteins to form a complex, thereby inhibiting *ARF’s* activation of downstream genes. In the presence of auxin, auxin binds to receptors such as TIR1, promoting the ubiquitination and degradation of AUX/IAA proteins. This releases the inhibition of *ARF* by AUX/IAA, enabling *ARF* to activate the expression of downstream auxin-responsive genes. This facilitates auxin signal transduction, thereby regulating plant growth and development processes such as cell enlargement and plant elongation [40]. The analysis of *cis*-acting elements within the promoter regions revealed that *ARFs* contained multiple light-responsive elements, hormone-responsive elements, *cis*-acting elements associated with plant growth and development, and *cis*-acting elements responding to stress. This finding indicated that *ARFs* are closely linked to plant growth and development. The variation in the types and numbers of *cis*-acting elements within *ARFs* suggests functional differentiation amongst these factors; numerous *cis*-acting elements within *PeARFs* are implicated in abiotic stress responses and plant growth and development, such as DRE *cis*-acting elements [41] and AREB *cis*-acting elements [42], thereby possibly ensuring the survival of *P. euphratica* in harsh arid environments.

### 3.3. Functional Characterisation of PeARF18

The expression patterns of *PeARFs* are primarily categorised into two groups: those up-regulated and down-regulated in Ov and Bo leaves compared with those in Li and La leaves. Multi-omics integration analysis and molecular regulatory network analysis were applied to identify *PeARFs* [43]. *PeARF18* (*PeuTF14G00975.1*) exhibited the highest expression in Ov and Bo leaves, with significant expression differences in Li and La. The heteromorphic leaves of *P. euphratica* are directly linked to its adaptation to arid environments. Given this pattern of expression, *PeARF18* may regulate the growth and development of *P. euphratica* leaves to adapt to arid environments. We also conducted interspecies collinearity analysis, revealing an expansion within the *PeARF* gene family. For instance, *PeARF18* exhibits collinearity with both *PeuTF02G01548.1* and *AtARF18*. However, its expression patterns and levels differ across various leaf morphs of *P. euphratica*, suggesting that functional shifts may have occurred within the *PeARF* gene family following this expansion.

## 4. Materials and Methods

### 4.1. Identification and Physicochemical Characterisation of ARF Family Members in Three Salicaceae Species

*PeARFs* were analysed on the basis of the *P. euphratica* genome [44]. The *ARFs* for the other two Salicaceae were determined on the basis of the genomic data from *P. pruinosa* (National Center for Biotechnology Information, Biological Project Accession Number PRJNA863418) and *S. sinopurpurea* [36]. All genomic data of the three poplars were firstly aligned with the protein sequences of the 23 *AtARF* genes’ family obtained from NCBI by using Blastp (https://blast.ncbi.nlm.nih.gov/Blast.cgi?PROGRAM=blastp&PAGE_TYPE=BlastSearch&LINK_LOC=blasthome, accessed on 5 December 2023) searches with an e-value of 1.0 × 10^−10^. In addition, searches were performed using HMMER for possible ARF proteins. Genes with complete CD length-specific ARF-conserved structural domains were then identified by further screening with Pfam batch sequence search (http://pfam.xfam.org/search, accessed on 12 December 2024) and NCBI batch CD-Search (https://www.ncbi.nlm.nih.gov/Structure/cdd/wrpsb.cgi, accessed on 12 December 2023). The candidate *ARF* genes were subsequently identified in the Salicaceae families by using SMART (http://smart.embl-heidelberg.de/, accessed on 13 December 2024). The isoelectric point of the PeARF proteins and the theoretical molecular weight (Mw) were predicted by ExPASy (https://www.expasy.org/, accessed on 14 December 2024). The subcellular localisation of the *ARF* families in Salicaceae was predicted by the online website WoLF PSORT (https://wolfpsort.hgc.jp/, accessed on 14 December 2024).

### 4.2. Phylogenetic Analysis of Multi-Species ARF Gene Family

The structural domains in the ARF protein sequences of *P. euphratica*, *S. sinopurpurea*, *P. pruinose*, and *A. thaliana* were retrieved using the SMART website. The sequences of ARF structural domains were extracted using the coordinates of the ARF structural domains and merged into a new sequence. The merged protein sequences were then used to construct a phylogenetic tree for the three species. The merged ARF protein sequences of *P. euphratica*, *S. sinopurpurea*, *P. pruinosa*, and *A. thaliana* were compared using the Clustal method of MEGA-X under default settings [45]. The neighbour-joining (NJ) method was used to infer the evolutionary history. A bootstrap consensus tree inferred from 1000 replicates was used to represent the evolutionary history of the analysed taxa. The percentage of replicate trees clustered by the relevant taxa in the bootstrap test (1000 replicates) is shown next to the branches. Evolutionary distances were calculated using a method on the basis of the Dayhoff matrix and expressed as the number of amino acid substitutions per site. All ambiguous positions were removed for each sequence pair (pairwise deletion option). The phylogenetic trees were landscaped and displayed using the ITOL online website (https://itol.embl.de/, accessed on 5 January 2025).

### 4.3. Analysis of Gene Structure and Conserved Structural Domains of ARF Family Members in Three Salicaceae Families

The conserved motifs were analysed using MEME (http://meme-suite.org/, accessed on 10 January 2025), with optimal widths ranging from 10 to 150. The number of motifs was set to 10, and the rest were set as defaults. Then, TBtools (v2.097) was used to plot the gene structures and motifs, and the results were analysed.

### 4.4. Analysis of Cis-Acting Elements of ARF Family Members in Three Salicaceae Plants

A total of 2000 bp sequences upstream of the start codon of the Salicaceae *ARF* gene family members were entered into the PlantCare website (https://bioinformatics.psb.ugent.be/webtools/plantcare/html/, accessed on 12 March 2025) to predict the *cis*-elements in the promoters of each gene, which were then visualised using TBtools to analyse the promoter sequences in the ARF family genes of three Salicaceae family plants and predict their functions.

### 4.5. Multi-Species Covariance and Chromosomal Localisation of Three Salicaceae ARF Families

Homologous pairs of *P. euphratica* with three other species (*P. pruinose*, *Salix sinopurpurea,* and *A. thaliana*) were screened by BLASTP comparison. Then, common intervals between *P. euphratica* and these three other species were screened using TBtools software and visualised using visualisation software. Chromosome length information (Fasta Stats), ID and position information of *PeARFs* genes (GFF3 gene position parse/Text Block Extract and Filter), and gene density information (Gene Density Profile) were extracted from the genome files of the three poplars using TBtools software. Gene Density Profile), and then the TBtools/Gene Location Visualize function was used to visualise the chromosome location.

### 4.6. Analysis of Expression Patterns

Thirty leaf samples of *P. euphratica* were collected from the *P. euphratica* forest in Alar City, Xinjiang Uygur Autonomous Region, China, with the specific location recorded as 40°32′36.90′′ N (latitude) and 81°17′56.52′′ E (longitude). From late July to August, leaves at the Li (linear leaves), La (lanceolate leaves), Ov (ovate leaves), and Bo (broadly ovate leaves) developmental stages of *P. euphratica* were sampled. Specifically, the third or fourth leaf segments from the base of current-year branches were selected as experimental materials. Upon collection, the samples were immediately stored at −80 °C in a freezer. Three biological replicates were prepared for transcriptome sequencing. Whole-transcriptome mRNA-seq was conducted on an Illumina HiSeq X-Ten platform (Novogene, Beijing, China), following the manufacturer’s standard protocol. Clean reads were mapped to the reference genome using HISAT2 (version 2.0.4) to quantify the expression levels of annotated *P. euphratica* genes [46]. Gene expression abundance was calculated as fragments per kilobase of transcript per million mapped reads (FPKMs) via StringTie (version 2.2.1) [47].

The heteromorphic foliage was collected from different canopies and stored in an ultra-low-temperature refrigerator at −80 °C after being rapidly frozen with liquid nitrogen. An actin gene was used as the endogenous control. Each reaction was performed in biological triplicates, and the CT values obtained through qRT-PCR were analysed using the 2^−∆∆CT^ method to calculate the relative fold change values.

## 5. Conclusions

This study provides the first comprehensive identification and analysis of *ARF* gene families in three species of Salicaceae. Their structure, phylogenetic relationships, *cis*-acting elements, colinearity, and subcellular localisation were analysed. This gene family may play a role in the hormone regulation of *P. euphratica* leaf development. RNA sequencing (RNA-seq) combined with quantitative reverse transcription polymerase chain reaction (qRT-PCR) revealed that the transcriptional dynamics of *PeARF18* strongly correlate with the developmental process of heteromorphic leaves in *P. euphratica*. A high expression of *PeARF18* in wild-type *P. euphratica* resulted in increased leaf area, consistent with its high expression in broad leaves and low expression in narrow leaves of *P. euphratica*. Therefore, *PeARF18* may promote the increase in leaf area of *P. euphratica*. In summary, this study provides a foundation for investigating the *ARF* gene family in *P. euphratica* and offers theoretical support on how *PeARF18* influences the growth and development of *P. euphratica*’s heteromorphic leaves. This study focuses on bioinformatics analysis and lacks functional validation through mutant or overexpressing plants; subsequent work will incorporate CRISPR editing technology or gene overexpression in poplars.

## Figures and Tables

**Figure 1 ijms-27-00335-f001:**
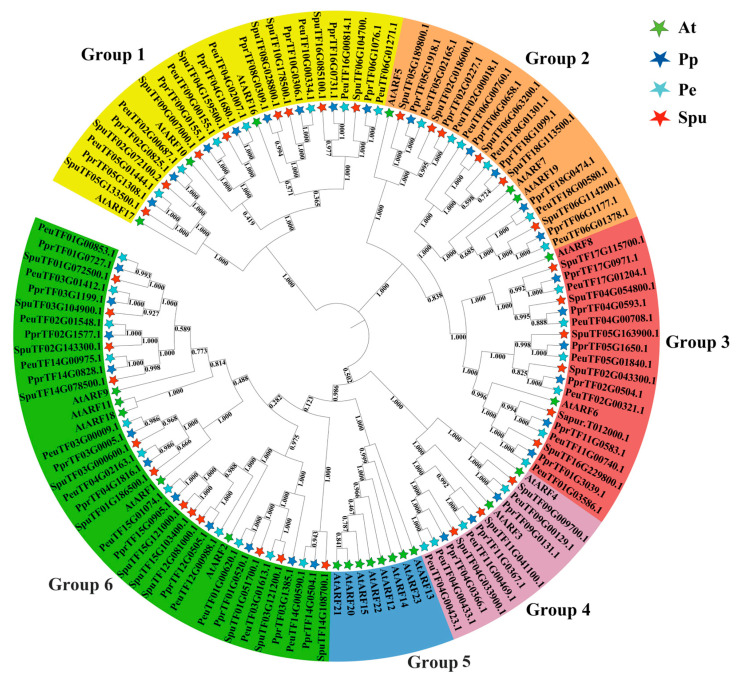
Neighbour-joining (NJ) tree of *ARF* gene family members in four species. Differently coloured arcs indicated various subgroups. Five-pointed stars of different colours represent different species. The full names of species are as follows: Pe = *P. euphratica*, At = *A. thaliana*, Pp = *P. pruinosa*, Spu = *Salix sinopurpurea*.

**Figure 2 ijms-27-00335-f002:**
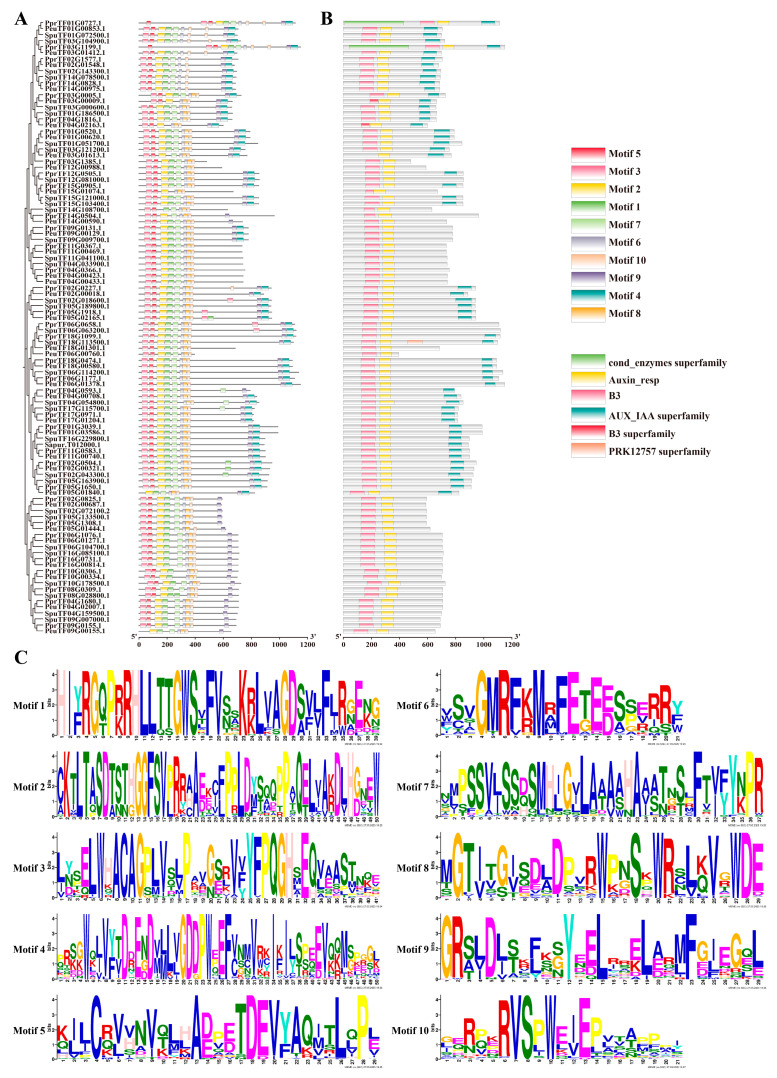
Conserved motifs and conserved structure analysis of *ARF* genes. (**A**) Evolutionary relationship and conserved motif. (**B**) The distribution of conserved structural domains of *PeARF* genes. (**C**) Conserved motif analysis of PeARF proteins.

**Figure 3 ijms-27-00335-f003:**
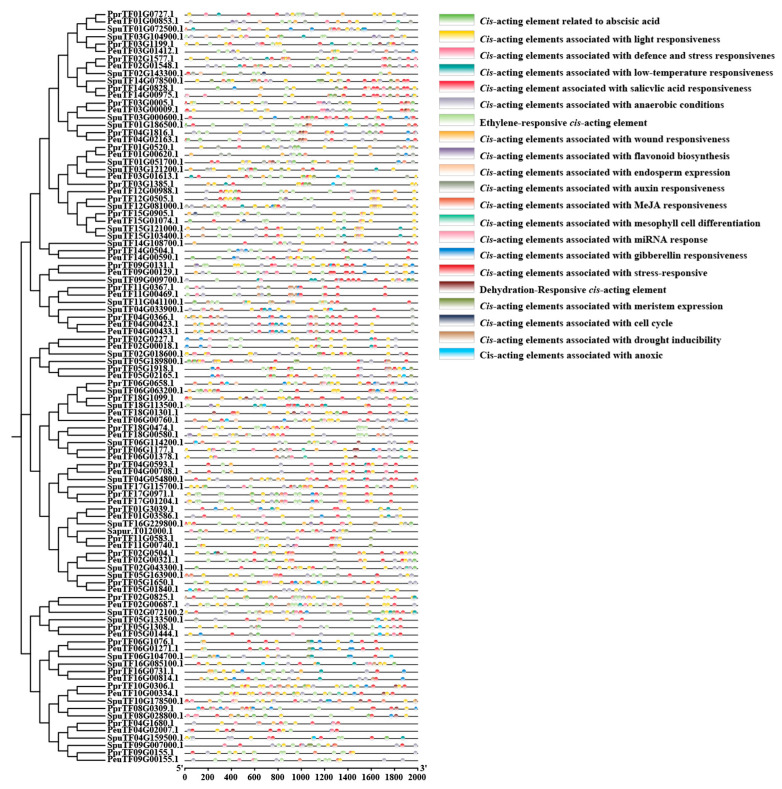
Putative *cis*-acting regulatory elements in the promoters of *ARF* genes. The *cis*-acting elements were analysed in the 2000 bp upstream promoter regions of corresponding *ARF* genes using the PlantCARE database. The rectangles of different colours represent distinct *cis*-acting elements.

**Figure 4 ijms-27-00335-f004:**
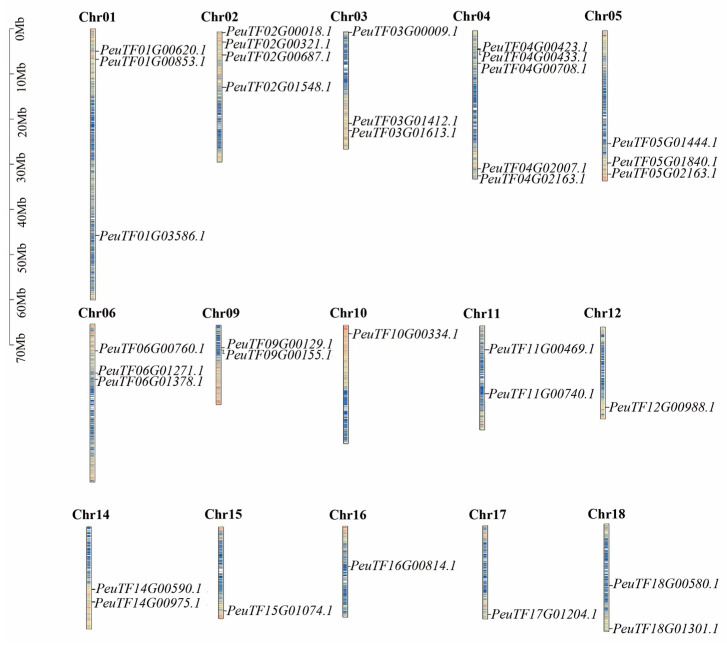
Analysis of chromosomal localisation of *PeARFs*. Blue indicates low gene density, while red indicates high gene density on the chromosome.

**Figure 5 ijms-27-00335-f005:**
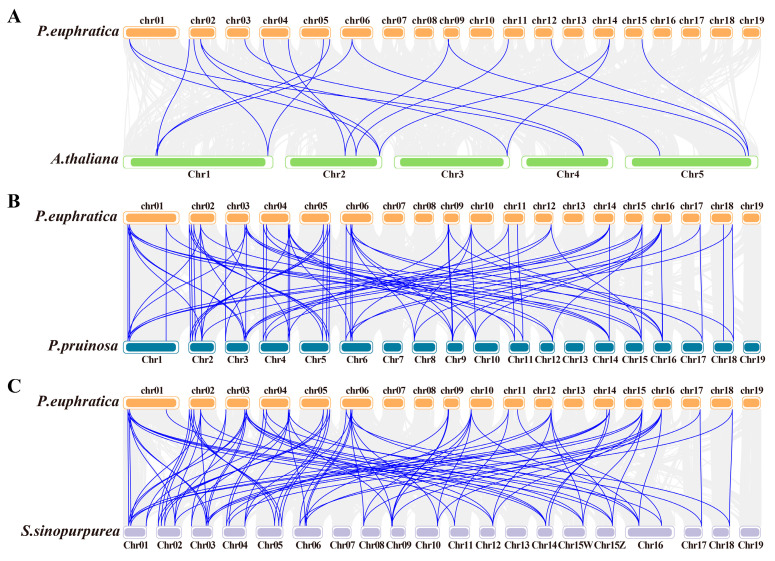
Collinearity analysis indicates that *P. europratica* shares collinearity with three other species: (**A**) *A. thaliana*, (**B**) *P. pruinose*, and (**C**) *S. sinopurpurea*. The grey lines in the background indicate collinearity groups within the *P. europratica* and other plant genomes, while the blue lines highlight collinear *ARF* pairs.

**Figure 6 ijms-27-00335-f006:**
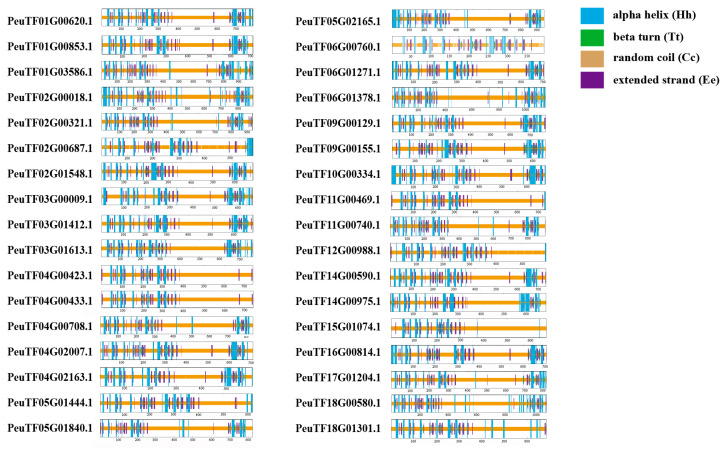
Secondary structure prediction of PeARF proteins. Hh (alpha helix), Tt (beta turn), Cc (random coil), and Ee (extended strand) are shown with different colours.

**Figure 7 ijms-27-00335-f007:**
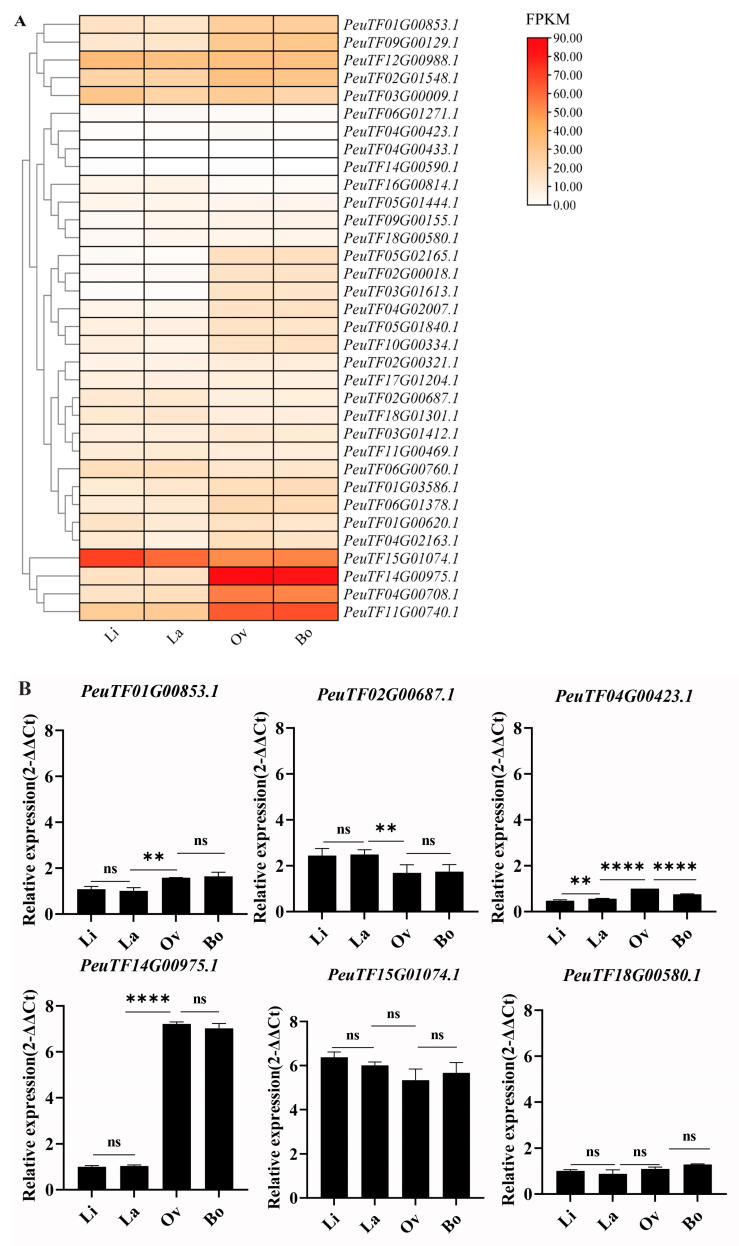
Expression patterns of PeARFs in heteromorphic leaves. Linear leaves, lanceolate leaves, ovate leaves, and broadly ovate leaves are abbreviated as Li, La, Ov, and Bo, respectively. (**A**) Expression profiles of PeARFs during leaf development. (**B**) qRT-PCR data for PeARF expression profiles in heteromorphic leaves. The ‘****’ indicates significant differences of *p* < 0.0001; the ‘**’ indicates significant differences of *p* < 0.01; ‘ns’ indicates no significance.

**Figure 8 ijms-27-00335-f008:**
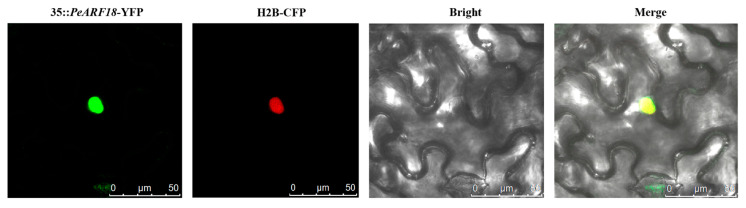
Nuclear localisation of the 35S::*PeARF18*-YFP protein in tobacco leaf epidermal cells: fluorescence images of PeARF18 (35S::*PeARF18*-YFP), nuclear localisation signals (H2B-CFP), and fusion images (35S::*PeARF18*-YFP/H2B-CFP). Scale bar = 25 µm.

## Data Availability

The RNA-seq data for *P. euphratica*’s heteromorphic leaves used in this study have been submitted to the National Genomics Data Center (https://ngdc.cncb.ac.cn/) under BioProject accession number PRJCA005959 (accessed on 13 January 2024).

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
