# Peer review of "Int. J. Mol. Sci.2026, 27(1), 335;https://doi.org/10.3390/ijms27010335"

_ijms, 2025, doi:10.3390/ijms27010335_

Round 1

Reviewer 1 Report

Comments and Suggestions for Authors

Dear Authors,
This manuscript presents one of many studies devoted to the identification of a specific gene family in plants. The manuscript requires significant revision to be accepted for publication in IJMS.

The abstract should be significantly shorter; about 200 words maximum, please see author guidelines (https://www.mdpi.com/journal/ijms/instructions).

Keywords must be unique and do not duplicate the title of the article. Please revise them.

The introduction is well-written, but I would ask the authors to try to answer the question of what makes their study stand out from similar ones, and to add a thought about this in the last paragraph.

Results
2.1. All gene names and their identifiers must be written in italics. Please correct this throughout the text.
Сould you provide all identified sequences in FASTA format in the supplementary file?
2.2. The labels for the identifiers in the phylogenetic tree in Figure 1 are almost unreadable and should be enlarged, as should the labels for the asterisks representing the different species.
2.3. Unfortunately, subplots A and B in Figure 2 are unreadable, and I advise you to upload them in full size to the supplement.
Line 160: "auxin-resp" → auxin-responsive; please check the entire text and remove such abbreviations if they are present in other places.
2.4. Figure 3 contains a very small and unreadable color legend, it needs to be enlarged.
Also, the text for all figures must be self-explanatory, and all abbreviations used in them must be explained.
2.5. Figure 4 with chromosomal localization is also unreadable. The figure captions need to be enlarged. Also, please add the color scale explanations used for chromosomes.
2.6. Also, the chromosome signatures are very small and should be enlarged in Figure 5.
2.7. I don't like the idea of combining the data table with the structures in Figure 6. I suggest you put all the table information about the sequences in the supplement, and provide the structures in high resolution and larger size, with a color legend and appropriate figure captions.
2.8. Figure 7A appears unnaturally stretched. Please restore its original dimensions and color legend. Also, please explain the units (counts-per-million, FPKM, or other) used to represent these expression values.
Figure 7B: The subplots need to be enlarged and scaled to a single Y-limit so they can be compared with each other.
Could you provide the correlation value between transcriptome data and qRT-PCR data?
2.9. I don't understand why this section is needed in the paper. It's a simple description of protein-protein interactions for PeARF18 from the STRING database; it doesn't add any new "predictions."
2.10. The size scale on the "Bright" and "Merge" charts is unreadable due to its white color.

Discussion 
The discussion contains excessive information at the end of the first paragraph of section 3.1, which can be shortened. Also, in section 3.3, the information about the protein-protein interactions "observed" in your experiment is incorrect. You did not reveal any novel interactions using STRING-db; if you, for example, had constructed coexpression networks, you could at least have made some claims about potential interactions between ARFs and targets, but here you simply cite publicly available and known information from the database. Taken together, the discussion feels more like a continuation of a literature review than an attempt to interpret the obtained data in the context of existing data.

Materials and methods
4.1. Please, provide links to all genomic data or their identifiers that you used. For each program, if applicable, provide its launch parameters.
4.5. What BLAST thresholds for comparison were applied?
4.7. In what year were the samples collected, and from what age were trees? How many trees were included in the experiment? "Li, La, Ov, and Bo" must be provided in full. Also, you haven't provided the bioinformatics processing protocol for the obtained transcriptome data. 
There's also no mention of qRT-PCR validation in the methods; it needs to be added separately.

Author Response

Comments 1: [The abstract should be significantly shorter; about 200 words maximum.]

Response 1: Thank you for pointing that out. We have adjusted the word count to under 200 words.

Comments 2: [Keywords must be unique and do not duplicate the title of the article. Please revise them.]

Response 2: Thank you for your valuable comments. The keyword section has been amended in line 30.

Comments 3: [The introduction is well-written, but I would ask the authors to try to answer the question of what makes their study stand out from similar ones, and to add a thought about this in the last paragraph.]

Response 3: Thank you for your valuable comments. We have amended that section in lines 81-88 on page 2.

Comments 4: [2.1. All gene names and their identifiers must be written in italics. Please correct this throughout the text. Could you provide all identified sequences in FASTA format in the supplementary file?]

Response 4: Thank you for your valuable comments. The revisions have been incorporated into our resubmitted manuscript. We have provided the FASTA format file for the identified sequences in the supplementary document.

Comments 5: [2.2. The labels for the identifiers in the phylogenetic tree in Figure 1 are almost unreadable and should be enlarged, as should the labels for the asterisks representing the different species.]

Response 5: Thank you for your valuable comments, and we have revised the figure in the resubmitted manuscript (line 141 on page 4).

Comments 6: [2.3. Unfortunately, subplots A and B in Figure 2 are unreadable, and I advise you to upload them in full size to the supplement.]

Response 6: Thank you for your valuable comments. They have been corrected in our resubmitted manuscript. We have increased the fonts of Figure 2 in page 5 (the image resolution is 600 dpi.).

Comments 7: Line 160: "auxin-resp" → auxin-responsive; please check the entire text and remove such abbreviations if they are present in other places.]

Response 7: The relevant issues have been revised in our resubmitted manuscript. For the specific revisions, please refer to the results ( lines 152, 154 on page 5 ).

Comments 8: [2.4. Figure 3 contains a very small and unreadable color legend, it needs to be enlarged. Also, the text for all figures must be self-explanatory, and all abbreviations used in them must be explained.]

Response 8: Thank you for your scrutiny. We have amended the image and provided an explanation in the caption (Figure3, lines 188-189 on page 6 ).

Comments 9: [2.5. Figure 4 with chromosomal localization is also unreadable. The figure captions need to be enlarged. Also, please add the color scale explanations used for chromosomes.]

Response 9: Thank you for your valuable comments. We have amended the resubmitted manuscript to include the colour scale explanations used for chromosomes (lines  208-209 on page 7 ).

Comments 10: [2.6. Also, the chromosome signatures are very small and should be enlarged in Figure 5.]

Response 10: Thank you for your review. We have made the necessary amendments in the resubmitted manuscript (line 220 on page 8 ).

Comments 11: [2.7. I don't like the idea of combining the data table with the structures in Figure 6. I suggest you put all the table information about the sequences in the supplement, and provide the structures in high resolution and larger size, with a color legend and appropriate figure captions.]

Response 11: Thank you for your valuable comments. We have incorporated the relevant form information into supplementary documentation as per your instructions (Table S3), and have included the images with high-resolution, large-format illustrations in the revised manuscript. Figure captions have been added accordingly (lines 242 and 244 on page 9).

Comments 12: [2.8. Figure 7A appears unnaturally stretched. Please restore its original dimensions and color legend. Also, please explain the units (counts-per-million, FPKM, or other) used to represent these expression values.

Figure 7B: The subplots need to be enlarged and scaled to a single Y-limit so they can be compared with each other.

Could you provide the correlation value between transcriptome data and qRT-PCR data?]

Response 12: Thank you for your valuable comments. We have amended the images and the unit for transcriptomic data has been supplemented: FPKM (Figure 7A). We have submitted the correlation coefficients for the transcriptomic data and qRT-PCR data in lines 255-257 of page 9.

Comments 13: [2.9. I don't understand why this section is needed in the paper. It's a simple description of protein-protein interactions for PeARF18 from the STRING database; it doesn't add any new "predictions."]

Response 13: Thank you for your valuable comments. We have deleted this content.

Comments 14: [2.10. The size scale on the "Bright" and "Merge" charts is unreadable due to its white color.]

Response 14: Thank you for your valuable comments. We have resubmitted the images in the revised manuscript (Figure 8, on page 11).

Comments 15: [The discussion contains excessive information at the end of the first paragraph of section 3.1, which can be shortened. Also, in section 3.3, the information about the protein-protein interactions "observed" in your experiment is incorrect. You did not reveal any novel interactions using STRING-db; if you, for example, had constructed coexpression networks, you could at least have made some claims about potential interactions between ARFs and targets, but here you simply cite publicly available and known information from the database. Taken together, the discussion feels more like a continuation of a literature review than an attempt to interpret the obtained data in the context of existing data.]

Response 15: Thank you for your valuable comments. We have already streamlined the concluding section of the first paragraph, and removed the section on protein interactions.

Comments 16: [4.1. Please, provide links to all genomic data or their identifiers that you used. For each program, if applicable, provide its launch parameters.]

Response 16: Thank you for your review. We have provided links or identifiers for the genomic data used on page 12, lines 354-357.

Comments 17: [4.5. What BLAST thresholds for comparison were applied?]

Response 17: Thank you for your scrutiny. The BLAST alignment threshold applied is the BLASTp website (e-value ≤ 1.0 × 10−10) on line 359 of page 13.

Comments 18: [4.7. In what year were the samples collected, and from what age were trees? How many trees were included in the experiment? "Li, La, Ov, and Bo" must be provided in full. Also, you haven't provided the bioinformatics processing protocol for the obtained transcriptome data. There's also no mention of qRT-PCR validation in the methods; it needs to be added separately.]

Response 18: Thank you for your scrutiny. We have supplemented the Materials and Methods section in our resubmitted manuscript (lines 412-430 on page 14).

Reviewer 2 Report

Comments and Suggestions for Authors

Dear Editor

Thank you for providing this opportunity to review the manuscript entitled “Genome‑Wide Identification of ARF Gene Family in Three Salicaceae Species and Functional Analysis of PeARF18 in Regulating Heteromorphic Leaves of Populus euphratica”. After careful observations, I have following minor comments for authors which need to be considered for this Journal.    

Comments for authors

Title

The title is overcrowded and overly descriptive, trying to convey too many elements comparative genomics, three species, expression profiling, heteromorphic leaf development, and PeARF18 characterization. A title should communicate the central contribution without listing every analysis.

Abstract

The abstract is densely packed with excessive numerical details, including motif counts, domain descriptions, duplication numbers, and structural predictions. These details overwhelm the key findings and reduce clarity.

Several sentences read like condensed Materials & Methods, which is not appropriate for an abstract.

The abstract makes overstated functional claims regarding PeARF18 (e.g., “validated” or “verified role”), but the manuscript presents only correlative evidence (expression patterns, predicted interactions, localization). These statements should be toned down to avoid implying experimental validation.

I recommend restructuring the abstract into 4 clear components: background, objective, main findings, conclusion/implication, while keeping it concise and high-level.

Keywords

The present list contains overlapping and overly broad terms. Several keywords repeat concepts already stated in the title or abstract. Keywords should be specific, non-redundant, and improve searchability rather than restate obvious terms.

Introduction

The background on ARF genes is appropriate, but the introduction lacks a focused rationale for studying three Salicaceae species together. Provide explicit justification why P. euphratica, P. pruinosa, and S. sinopurpurea were selected (ecosystem relevance, evolutionary divergence, leaf morphology differences).

The concept of heteromorphic leaves is introduced but without detailing how ARFs are hypothesized to regulate this trait based on prior studies. Add a concise summary of known ARF roles in leaf polarity, lamina expansion, or auxin-dependent leaf shaping.

The Introduction does not explicitly define the knowledge gap your study aims to fill. Please insert a dedicated paragraph clearly defining the scientific gap and explaining why previous studies are insufficient.

The final paragraph of the Introduction includes statements that closely resemble summaries of your results, rather than motivations for conducting the study.

Statements such as describing what you “found”, “revealed”, or “identified” should not appear in the Introduction.

Please revise this paragraph so that it focuses on: the rationale for the study, the knowledge gap, what the study intends to investigate, and remove any hints of your conclusions.

Materials and methods

Add reference for MEGA-X.

In phylogenetic analysis, indicate the alignment tool.

The PlantCARE link contains a formatting error; fix the URL.

Results

For genes lacking B3 or AUX/IAA domains, provide sequence evidence to confirm truncation rather than annotation errors.

Improve the resolution of figure 2A, 2B, 4 & 8, as these are partially blurred.

The cross-species collinearity analysis is well presented, but anchor gene lists must be provided as Supplementary files.

qRT-PCR figure (Figure 7B) lacks p-value annotations and standard error definitions.

In figure 9, scale bar is missing for “Merge”, not visible for “Bright”

Discussion

The Discussion often repeats Results without deep mechanistic interpretation.

Overstatements: several sentences claim that PeARF18 “regulates” heteromorphic leaf development; revise to “likely involved” based on expression correlation.

Integrate RNA-seq and synteny findings more clearly for example, whether duplicated ARFs show neofunctionalization.

Conclusion

The Conclusion restates results rather than synthesizing them.

Remove claims of functional verification of PeARF18 unless genetic manipulation experiments are added.

Provide a forward-looking statement identifying next steps (functional assays, mutant analyses).

Author Response

Comments 1: [The title is overcrowded and overly descriptive, trying to convey too many elements comparative genomics, three species, expression profiling, heteromorphic leaf development, and PeARF18 characterization. A title should communicate the central contribution without listing every analysis.]

Response 1: Thank you for your valuable comments. We have replaced the primitive title with “Characterization of the ARF Gene Family in Salicaceae and Functional Analysis of PeARF18 in Heteromorphic Leaf Development of Populus euphratica “ in the resubmitted manuscript.

Comments 2: [The abstract is densely packed with excessive numerical details, including motif counts, domain descriptions, duplication numbers, and structural predictions. These details overwhelm the key findings and reduce clarity.]

Response 2: Thank you for your valuable comments. We have removed unnecessary numerical details from the abstract and streamlined the summary section in the resubmitted manuscript.

Comments 3: [Several sentences read like condensed Materials & Methods, which is not appropriate for an abstract.]

Response 3: Thank you for your valuable comments. We have revised the abstract in the resubmitted manuscript.

Comments 4: [The abstract makes overstated functional claims regarding PeARF18 (e.g., “validated” or “verified role”), but the manuscript presents only correlative evidence (expression patterns, predicted interactions, localization). These statements should be toned down to avoid implying experimental validation.]

Response 4: Thank you for your valuable comments. The overstated functional claims regarding PeARF18 were deleted in the abstract.

Comments 5: [I recommend restructuring the abstract into 4 clear components: background, objective, main findings, conclusion/implication, while keeping it concise and high-level.]

Response 5: Thank you for your valuable comments. We have revised the abstract section in accordance with your suggestions and divided it into the following sections: background, objective, main findings, and conclusion/implications.

Comments 6: [The present list contains overlapping and overly broad terms. Several keywords repeat concepts already stated in the title or abstract. Keywords should be specific, non-redundant, and improve searchability rather than restate obvious terms.]

Response 6: Thank you for your valuable comments. The keyword section has been amended in line 30.

Comments 7: [The background on ARF genes is appropriate, but the introduction lacks a focused rationale for studying three Salicaceae species together. Provide explicit justification why P. euphratica, P. pruinosa, and S. sinopurpurea were selected (ecosystem relevance, evolutionary divergence, leaf morphology differences).]

Response 7: Thank you for your comments. We have described the differences in leaf morphology among these three drought-tolerant Salicaceae species in lines 75–88 of the introduction.

Comments 8: [The concept of heteromorphic leaves is introduced but without detailing how ARFs are hypothesized to regulate this trait based on prior studies. Add a concise summary of known ARF roles in leaf polarity, lamina expansion, or auxin-dependent leaf shaping.]

Response 8: Thank you for your valuable comments. We have cited the related references on the role of ARFs in regulation of leaf shape on page 2, lines 63–74 and 78–81.

Comments 9: [The Introduction does not explicitly define the knowledge gap your study aims to fill. Please insert a dedicated paragraph clearly defining the scientific gap and explaining why previous studies are insufficient.]

Response 9: Thank you for your review. We have made revisions and additions to the final paragraph of the Introduction in lines 81–88.

Comments 10: [The final paragraph of the Introduction includes statements that closely resemble summaries of your results, rather than motivations for conducting the study.]

Response 10: Thank you for your valuable comments. We have made amendments and additions to lines 89-100 in accordance with your suggestions regarding the research rationale.

Comments 11: [Statements such as describing what you “found”, “revealed”, or “identified” should not appear in the Introduction.]

Response 11: Thank you for your valuable comments. The overstated functional claims regarding PeARF18 were deleted in the introduction.

Comments 12: [Please revise this paragraph so that it focuses on: the rationale for the study, the knowledge gap, what the study intends to investigate, and remove any hints of your conclusions.]

Response 12: Thank you for your valuable comments, we have revised our introduction according to your suggestion.

Comments 13: [Add reference for MEGA-X.

In phylogenetic analysis, indicate the alignment tool.

The PlantCARE link contains a formatting error; fix the URL.]

Response 13: Thank you for your valuable comments. We have made amendments in accordance with the above three recommendations on page 13-14, lines 378-379, 397.

Comments 14: [For genes lacking B3 or AUX/IAA domains, provide sequence evidence to confirm truncation rather than annotation errors.]

Response 14: Thank you for your valuable comments. We have added annotations to lines 156-157 of the manuscript. We have provided multiple sequence alignment results (Figure S1) in the supplementary materials.

Comments 15: [Improve the resolution of figure 2A, 2B, 4 & 8, as these are partially blurred.]

Response 15: Thank you for your valuable comments. We have amended these images in the resubmitted manuscript. Figure 8 has been removed as it lacks persuasive value.

Comments 16: [The cross-species collinearity analysis is well presented, but anchor gene lists must be provided as Supplementary files.]

Response 16: Thank you for your valuable comments. We have added annotations to lines 217 of the manuscript and have submitted this supplement in the supplementary materials (Table S2).

Comments 17: [qRT-PCR figure (Figure 7B) lacks p-value annotations and standard error definitions.]

Response 17: Thank you for your scrutiny. In the resubmitted manuscript, this information has been supplemented in the figure caption for Figure 7B on pages 11, lines 274-275.

Comments 18: [In figure 9, scale bar is missing for “Merge”, not visible for “Bright”]

Response 18: Thank you for your scrutiny. We have revised it in our resubmitted manuscript.

Comments 19: [The Discussion often repeats Results without deep mechanistic interpretation.]

Response 19: Thank you for your scrutiny. We have revised the discussion section and provided a more in-depth interpretation of the content in lines 288-350.

Comments 20: [Overstatements: several sentences claim that PeARF18 “regulates” heteromorphic leaf development; revise to “likely involved” based on expression correlation.]

Response 20: Thank you for your scrutiny. We have revised the content in our resubmitted manuscript.

Comments 21: [Integrate RNA-seq and synteny findings more clearly for example, whether duplicated ARFs show neofunctionalization.]

Response 21: Thank you for your scrutiny. Our analysis combining interspecies colinearity and RNA-seq data revealed that in P. euphratica, PeARF18 and PeuTF02G01548.1 share colinearity with AtARF18, but their functions may have diverged. We have added the description on page 12, lines 345-350.

Comments 22: [The Conclusion restates results rather than synthesizing them.]

Response 22: Thank you for your scrutiny. The conclusion was revised in our resubmitted manuscript on page 14-15, lines 431-446.

Comments 23: [Remove claims of functional verification of PeARF18 unless genetic manipulation experiments are added.]

Response 23: Thank you for your scrutiny. We have amended the relevant content in the conclusion section.

Comments 24: [Provide a forward-looking statement identifying next steps (functional assays, mutant analyses).]

Response 24: Thank you for your scrutiny. We have provided the next objectives in the resubmitted manuscript on page 15, lines 444-446.

Round 2

Reviewer 1 Report

Comments and Suggestions for Authors

Dear Authors,

After revision your manuscript can be accepted for publication in IJMS